# Influence of Pre-Tension on Free-End Torsion Behavior and Mechanical Properties of an Extruded Magnesium Alloy

**DOI:** 10.3390/ma16155343

**Published:** 2023-07-29

**Authors:** Hongbing Chen, Zhikang Shen, Bo Song, Jia She

**Affiliations:** 1College of Engineering and Technology, Southwest University, Chongqing 400715, China; zhikangshen@swu.edu.cn; 2School of Materials and Energy, Southwest University, Chongqing 400715, China; bosong@swu.edu.cn; 3College of Materials Science and Engineering, Chongqing University, Chongqing 400044, China; jiashe@foxmail.com

**Keywords:** magnesium alloys, free-end torsion, pre-tension, dislocations strengthening, mechanical properties

## Abstract

In this study, the influence of pre-tension on free-end torsion behavior and compression mechanical properties and micro-hardness of an extruded AZ31 Mg alloy was investigated using electron backscatter diffraction (EBSD), compression testing and micro-hardness testing. The result indicates that pre-tension can cause significant dislocation strengthening, which can increase the torsion yield strength and make the shear stress–shear strain curve of the pre-tension sample almost parallel to that of the as-extruded sample during plastic deformation stage. Texture in edge position on the cross-section of both the pre-tension and as-extruded samples can be rotated towards the extrusion direction by about ~30° by free-end torsion. The Swift effect is mainly responsible for the occurrence of massive extension twins in the central region. In contrast, normal stress is the main cause of extension twins occurring in the edge region. However, the effect of extension twins on micro-hardness is less than that of dislocations. The micro-hardness of both free-end torsion specimens increases almost linearly with increasing distance from center to edge on the cross-section. Nevertheless, the increase in micro-hardness of the pre-tension and then torsion sample is inconspicuous because pre-tension leads to dislocation proliferation and dislocation accumulation saturation. The result also indicates that both pre-tension and free-end torsion can lead to dislocation strengthening, which can obviously increase the micro-hardness and compressive yield stress. The underlying mechanisms were explored and discussed in detail.

## 1. Introduction

Magnesium alloys are the lightest of all the engineering metals, about two-thirds the density of aluminum and one-quarter the density of steel, and are plentiful on Earth. It is considered to be the most promising structural metal for automobile, aerospace and 3C (computer, consumer electronics and communication) products due to its excellent properties such as low density, high specific strength, good stiffness and superior electromagnetic shielding [1,2,3]. However, most Mg alloys inherently have low strength and poor formability at room temperature, which has prevented their widespread use in industry. A great deal of research has been devoted to improving the strength of Mg alloys over the last few decades. For instance, dislocation strengthening, solution strengthening, age hardening, grain refinement strengthening and deformation twinning strengthening are the conventional process methods for strengthening Mg alloys [4,5,6].

For most Mg alloy products, the forming processes involve strain path changes, i.e., pre-strain followed by further plastic deformation. It is therefore necessary to understand the pre-strain involved and how it affects Mg alloys during the manufacturing process. It is extensively accepted that pre-strain can heavily influence the active deformation mechanisms and deformation behavior in metallic materials [7]. Pre-strain, which can lead to dislocation strengthening and grain refinement strengthening by means of twin boundary subdivision, has been increasingly considered for strengthening Mg alloys due to its low cost and convenience [6]. Recently, extensive research has focused on the effect of pre-compression and pre-tension on subsequent deformation behavior [8,9,10]. Guo et al. [8] reported that pre-tension associated with annealing treatment can effectively increase the yield stress of the AZ31 Mg alloys plate. Yang et al. [11] found that a pre-compression strain level of 3.5% could significantly improve the torsion yield strength of the extruded AZ31 Mg alloy. This is mainly attributed to the grain refinement strengthening induced by extension twin boundary subdivision. Wang et al. [12,13] reported that the combination of pre-tension and torsion sequential deformation can refine grains more effectively than single torsion deformation for pure copper. Chen et al. [14] have shown that pre-tension can have a major effect on microstructure evolution and mechanical properties of pure titanium during a combination of pre-tension and torsion deformation.

Recently, pre-torsion deformation has been shown to be an effective method of improving the compressive properties of magnesium alloys [6]. The microstructure evolution during torsion is a critical factor influencing the compression property. Yang et al. [11] reported that pre-compression deformation can influence the torsion property and microstructure evolution during torsion due to the formation of extension twins. It is well known that tension and compression result in very different microstructure evolution [8,10]. In addition, magnesium alloy products are often subjected to a variety of loads that coexist in practical applications, such as combined tensile–torsional deformation, combined bending–torsional deformation and so on. However, the effect of pre-tension on free-end torsion behavior (i.e., combined tensile–torsional deformation) of extruded AZ31 Mg alloys is not well understood. The main objective of this present paper is to explore the impact of pre-tension on free-end torsion behavior for an extruded AZ31 Mg alloy, i.e., the AZ31 rods were first subjected to pre-tension, and then unloaded and free-end torsion was then carried out. The microstructure evolution during pre-tension and free-end torsion was investigated, and the influence of pre-tension and free-end torsion on micro-hardness and compressive yield strength were discussed in detail.

## 2. Materials and Methods

The schematic diagram of the current experiment is shown in Figure 1. To investigate the effect of pre-tension on free-end torsion deformation, two types of specimens were prepared. The raw material of AZ31 alloy (Mg-3 wt.%Al-1 wt.%Zn) was hot extruded at 350 °C into a rod with a diameter of 16 mm, at a ram speed of 1 m/min and an extrusion ratio of 25:1. After extrusion, the Mg alloy bar was air-cooled. The specimens were made from an extruded rod with the axis parallel to the extrusion direction (ED). One part of the initial sample was pre-stretched to a strain level of 10%, known as the pre-tension sample. The other part of the initial sample was subjected to re-turning, i.e., second turning, known as the as-extruded sample. The purpose of re-turning is to ensure that the diameter and gauge length of the sample is equal to that of the pre-tension sample. Subsequently, both specimens were subjected to free-end torsion deformation under the same conditions at room temperature. Finally, the two types of specimens were subjected to compression tests and micro-hardness tests to examine the effect of pre-tension on free-end torsion. Each test was repeated three times in order to obtain the most convincing results.

The geometry and dimensions of the dog-bone-shaped sample are shown in Figure 2a. Pre-tension deformation with a plastic strain level of 10% was performed on cylindrical dog-bone-shaped specimens (φ10 × 70 mm). For most extruded AZ31 Mg alloys, the tensile plastic strain can be up to 15–20% before rupture [6,15]. A pre-tension strain level of 10% was used to achieve adequate plastic deformation without necking in the current study. Finally, the gauge section of the initial sample was stretched to a length of 77 mm and shrunk to a diameter of φ9.5 mm, as shown in Figure 2b. To achieve the same geometry and dimensions, some of the initial specimens were re-turned to the dimensions of φ9.5 × 77 mm. The pre-tension testes were carried out on the universal test machine (LD26.105, Li Shi company, Shanghai, China) at a constant strain rate of 1 × 10^−3^/s. Free-end torsion tests were executed on a torsion test machine (NDW30500, Ke Xin company, Changchun, China) at a constant rate of 180°/min. Each end of the sample was fixed to grips with the help of a three-jaw chuck. For more experimental details, please refer to our previous literature [15]. For free-end torsion, it is supposed that there is no axial constrain, and the outside surface of the sample is traction-free [16]. After free-end torsion, the samples remained axisymmetric and homogeneous along the ED. In this study, the pre-tension sample and as-extruded sample can be twisted up to 420° and 500° before fracture, respectively. In order to avoid the risk of fracture, both types of specimens were designed to be twisted up to 290° under the same conditions at room temperature. The shear strain at the outer surface of the free-end torsion deformation rods is approximately 0.31. Some samples with the dimensions of φ9.5 × 15 mm were cut by wire electrical discharge machining for compression tests and micro-hardness tests. Compressive tests were performed on a universal test machine at a constant strain rate of 1 × 10^−3^/s at room temperature. The 0.2% proof stress was used to quantify the yield strength (YS). Each test was repeated three times to obtain representative results. Micro-hardness was measured using an MHV2000 hardness tester(Caikang company, Shanghai, China). The applied load was 1000 gf and the dwell time was 15 s. Each hardness value was tested three times and then the average value was calculated. Microstructural observation was carried out by scanning electron microscopy (JSM-6610, Japan Electronics company, Tokyo, Japan) using an accelerating voltage of 20 kV. Electron backscatter diffraction (EBSD, Oxford AZtech Max2) analysis was carried to identify the twins on a JSM-6610 scanning electron microscope equipped with an Oxford-EBSD system with a step size of 1 µm. An HKL Channel 5 System (Oxford system equipped in a JSM-6610) was applied to analyze the EBSD data. The EBSD specimens were cut from the middle part of the gauge section. EBSD measurements were carried out on the longitudinal section of the specimens, characterizing regions close to the surface and the core of the rods, i.e., edge position and center position, respectively. EBSD specimens were ground mechanically followed by electrochemical polishing in a commercial AC2 electrolyte for 30 s at 20 V. The characteristics of the differently orientated grains were observed using the inverse pole figure map (IPF). The evolution of the grain boundary and twin boundary was studied using grain boundary maps (GB). In order to assess the dislocation density, the kernel average misorientation (KAM) value was calculated. In general, KAM has been widely accepted and used as an indication of dislocation density, referring to the mean misorientation between a given point and its nearest neighbor within the same grain [17,18].

## 3. Results and Discussion

### 3.1. Shear Stress–Shear Strain Curves

In order to assess the influence of pre-tension on free-end torsion deformation behavior, both the as-extruded and pre-tension samples were subjected to free-end torsion deformation under the same conditions. Figure 3a illustrates the torque–twist angle curves for the two types of samples. Using Formula (1) in ref. [16], it is possible to draw the shear stress–shear strain curves as shown in Figure 3b.
(1)$$\gamma =\frac{R_{0}}{L_{0}}\varphi \;\;\;\;\tau =\frac{3Τ}{2\pi R_{0}^{3}}$$
where *γ* is the shear strain at the outer surface of the rod, and *τ* is the normalized torque. *L*_0_ and *R*_0_ are the sample’s initial length and radius, respectively. The specimen is subjected to a twist, *φ*, giving rise to a torque, T. From Figure 3, it can be seen that the torsion yield strength of the pre-tension sample is distinctly higher than that of the as-extruded one. In addition, as the shear strain increases, the shear stress also increase due to work hardening.

In comparison with ref. [11], where extruded AZ31 Mg alloy samples were subjected to pre-compression with plastic strain level of 1.5%, 3.5%, 5.5% and 6.5%, the shear stress–shear strain curves of the two types of pre-deformed samples are clearly different, as shown in Figure 4. On the one hand, the shear stress–shear strain curve of the pre-tension sample in the current study is almost parallel to that of the as-extruded sample during the plastic deformation stage. However, as indicated by the yellow ellipse in Figure 4, the curve of the pre-compression sample is not always parallel to that of the as-extruded sample. On the other hand, the torsion yield strength of the pre-tension sample is also higher than that of the pre-compression sample. The torsion yield strength of the as-extruded sample is about 75 MPa in this study, which is in agreement with the reported values in the literature [11,15]. However, the torsion yield strength of the pre-tension sample is about 110 MPa, an increase of about 35 MPa and a percentage increase of 46.67%, while that of the pre-compression sample is about 100 MPa, an increase of about 25 MPa and a percentage increase of 33.33% when compared with that of the as-extruded sample. It is reasonable to infer that pre-tension has a more significant effect than pre-compression on free-end torsion deformation for extruded AZ31 Mg alloys.

It can be seen that free-end torsion deformation behavior is greatly affected by pre-deformation, as shown in Figure 4. It is well known that extension twinning dominates the compression deformation along the ED for extruded Mg alloy rods [11]. Whereas in extruded Mg alloy rods, since extension twins and basal slips are difficult to activate due to their unfavorable orientation, non-basal slips are the dominant deformation mechanism when tension along the ED [19]. Hence, it can be concluded that pre-existing extension twins and dislocation slips have different effects on free-end torsion deformation. To some extent, the dislocation slip has a more significant strengthening effect than the extension twinning for extruded AZ31 Mg alloys during free-end torsion deformation. This is because pre-compression can generate many extension twins, which can form the c-axis//ED texture component, and this crystallographic orientation is favorable for activation of prismatic slips in subsequent free-end torsion, enhancing multiple slips and dynamic recovery [11]. Consequently, extension twins can further increase plasticity, which will cause the shear stress–shear strain curve of the pre-compression sample to decline, i.e., the curve will show softening at later stages of plastic deformation. Yang et al. [11] reported that when the pre-strain level is greater than 3.5%, the increase in torsion yield strength of the pre-strained specimen is inconspicuous. As the pre-strain level is further increased, the increase in torsion yield strength is mainly attributed to dislocation hardening. Refs. [16,20] reported that both extension twins and dislocation slips can be activated during free-end torsion deformation for AZ31 Mg alloys. Pre-tension can motivate substantial dislocations, which will proliferate and accumulate, thus inhibiting new dislocation movement and twinning deformation during free-end torsion, which is the primary strengthening mechanism [7]. Therefore, a pre-tension sample with substantial pre-existing dislocations will require more activation stress to induce dislocation slips and extension twinning to accommodate plastic deformation. Since there is no significant grain refinement after tensile deformation, grain refinement strengthening can be negligible for a pre-tension sample during free-end torsional deformation. Consequently, the shear stress–strain curve is higher than that of the as-extruded one, while dislocation strengthening is the primary strengthening mechanism for the pre-tension sample during free-end torsion. While for the pre-compression sample, pre-compression along the ED can give rise to a great deal of extension twins, which can introduce the Hall–Petch effect due to grain refinement by extension twins lamellae dividing. Furthermore, a high compression strain level can also increase dislocation density, which can also strengthen the dislocation strengthening. Therefore, both dislocation strengthening and grain refinement strengthening by means of twin boundary subdivision contributed to the strengthening of the pre-compression sample during free-end torsion deformation.

### 3.2. Microstructure Evolution during Pre-Tension and Free-End Torsion

#### 3.2.1. Microstructure of the Initial Sample and Pre-Tension Sample

As indicated in Figure 5a, the as-extruded sample has a well-distributed microstructure, which consists of some fine grains close to equiaxed grains with a mean grain size of approximately 14.5 µm measured from the EBSD micrograph. It can be seen that the c-axes of most grains are almost perpendicular to the ED, which is a typical extrusion texture in Mg-Al alloys [6,11,21]. From the GB maps, it can be seen that the as-extruded sample is twin-free. Moreover, the as-extruded sample has a low KAM value (~0.65°). Since the pre-tension strain level of 10% is moderate, there is inconspicuous variation with respect to the morphology of individual grains compared to the initial one, as shown in Figure 5b. For extruded Mg alloys, it is well known that non-basal slip is the dominant deformation mechanism during tension deformation. Tensile deformation can generate a large number of dislocations and cause dislocation strengthening. As a result, the pre-tension sample has a high KAM value (~1.17°). In addition, a few contraction twins and extension twins appeared in the pre-tension sample.

It is well known that dislocation slip and twinning are the main deformation modes in Mg alloys during plastic deformation [22]. Their activation is intensely dependent on the orientation relationship between the loading direction and crystallographic orientation. Furthermore, the critical resolved shear stress (CRSS) is a significant parameter for activation of different deformation modes. Different dislocation slips and twinning have different CRSS values. For instance, the CRSS of the basal dislocation slip is approximately 0.5 MPa, the CRSS of the prismatic dislocation slip has been reported to be in the range of 30–50 MPa, and the pyramidal dislocation slip has an even higher CRSS of 30–80 MPa [23]. Additionally, The CRSS values for the activation of extension twinning and contraction twinning were 2–5 MPa and 30–100 MPa, respectively [24]. In this study, the c-axis of most grains for the as-extruded AZ31Mg alloy are perpendicular to the loading direction (i.e., ED) during tensile deformation. Although the loading direction is favorable for contraction twinning, the CRSS for the activation of contraction twinning is relatively high. As a result, only a few contraction twins were activated in the pre-tension sample. Although basal dislocation slips and extension twinning have a lower CRSS value, they are difficult to activate under tension along the ED using the Schmid factor (SF) analysis. To estimate which slip systems are favorable for activation during pre-tension deformation, the Schmid factors of different deformation modes were calculated. As shown in Figure 6, the Schmid factors of prismatic <a> and two types of pyramidal <c + a> slip systems are obviously higher than that of the basal <a> slip. The results of this study are consistent with those reported in the literature [18,23]. Recently, a few researchers have also carried out tensile experiments on AZ31 and Mg-Gd Mg alloys, and found that many non-basal dislocations appeared in the deformed samples by TEM observation [23,25]. Non-Schmid behavior of extension twinning can also activate a few of extension twins in the pre-tension sample [26,27]. This is mainly due to local stress fluctuation caused by local non-uniform strain or strain compatibility between adjacent grains [26]. Therefore, a non-basal dislocation slip is the main deformation mode in the pre-tension sample, which can produce dislocation pile up and dislocation hardening. Consequently, the pre-tension sample has a higher KAM value (~1.17°) than that of the as-extruded sample (~0.65°), as shown in Figure 5.

#### 3.2.2. Microstructure of the Free-End Torsion Sample

Free-end torsion deformation may result in a non-homogeneous microstructure due to the inherent shear strain gradient across the cross-section of the rod [21]. In order to reveal the microstructure evolution, the edge and center positions were characterized by EBSD as two representative areas. From Figure 7a,c, it can be seen that the texture in the edge position on the cross-section suffers a rotation for the two free-end torsion samples. It seems that the orientation has been rotated by an angle of about 30° along the ED compared with the initial extruded texture, as shown in Figure 5. The texture rotated in this study is identical to that reported in refs. [21,28]. Beausir et al. [29] indicated that the ideal orientation of texture for extruded Mg alloys during torsion deformation is the B-fiber texture with the c-axis//ED. In the current study, the shear strain (*γ* = 0.31) is too small to generate an ideal B-fiber texture. The texture in the center position of the two free-end torsion samples is almost identical to that of the as-extruded sample, as shown in Figure 7b,d. This is because that shear strain in the center position is too small to rotate the texture around the radial axis.

In general, in addition to shear stress, normal stress is also applied to the cross-section during free-end torsional deformation from a material mechanics perspective. In the case where mild steel and cast iron were subjected to free-end torsion, it is well known that the mild steel is broken by shear stress and cast iron by normal stress. This means that normal stress is perpendicular to the c-axis of most grains for an extruded Mg alloy, which can lead to extension along the c-axis. This is the reason why massive extension twins can be activated in the edge area on the cross-section during free-end torsion. It is worth noting that in the EBSD map, the amount of extension twin lamellae in the center position is greater than that in the edge position. Theoretically, the shear strain at the center is relatively small or even zero, which is insufficient to initiate extension twins. The Swift effect is mainly responsible for the extension twins in the center area, i.e., the second-order axial effect under free-end torsion [30]. The Swift effect, which refers to an irreversible change in length during monotonic free-end torsion of metallic specimens, is an unusual phenomenon first described by Swift (1947) [31]. This is because the extension twins induced by free-end torsion at the edge can trigger the Swift effect. The Swift effect can cause the entire gauge section to contract along the ED while expanding along the radius. Carneiro et al. [20] carried out free-end torsion deformation on a rolled AZ31Mg alloy, and the results indicated that the maximum axial strain level to fracture was up to 3.5%. Hu et al. [30] also carried out free-end torsional deformation on an extruded AZ31Mg alloy, and calculated the compressive strain of the PRT15 specimens (*γ* = 0.15) to be up to 0.62%, which exceeds the compressive strain required to produce extension twins. Cazacu et al. [32] also used an elastic–plastic model to simulate the Swift effect. The results indicated that for a given orientation, if the tensile yield stress is greater than the compressive yield stress, the specimen will shorten when twisted along that direction and vice versa. In this study, the tensile yield strength is higher than the compressive yield strength of the extruded AZ31 Mg alloys, which is in agreement with the values reported in refs. [15,33]. It is therefore reasonable to conclude that the extension twins that have occurred in the central region are mainly due to the Swift effect. Furthermore, the extension twin at the edge region seems to be unaffected by the Swift effect. The reason for this is that the earliest extension twin at the edge region is caused by normal stress (normal stress and shear stress coexist during free-end torsion) rather than axial contraction by the Swift effect.

From Figure 7, it can be seen that free-end torsion deformation leads to an increase in the KAM value. The KAM value of both free-end torsion samples is remarkably higher than that of the initial sample. Furthermore, the KAM value at the edge area is obviously higher than that at the center area, as shown in Figure 7a,c. This indicates that the edge area was subjected to severe plastic deformation, resulting in a high density of dislocations. In contrast, the KAM value of the pre-tension and then torsion sample is the highest than the others, as shown in Figure 7c.

### 3.3. Compressive Mechanical Properties

Uniaxial compression tests were performed to evaluate the influence of pre-tension and free-end torsion on the mechanical properties of the extruded AZ31Mg alloy. Figure 8 exhibits the representative compressive stress–strain curves of various samples. Compressive mechanical properties can be derived from the true stress–true strain curves as shown in Table 1. For the as-extruded sample, the compressive yield stress (CYS) and the peak strength (PS) are 110.8 MPa and 338.7 MPa, respectively. The CYS of the pre-tension sample, free-end torsion sample and pre-tension and then torsion sample are 174.9 MPa, 173.9 MPa and 206.8 MPa, respectively. In comparison with the as-extruded sample, the percentage increase in CYS of the three samples is approximately 57.85%, 56.95% and 86.64%, respectively. It can be seen that the CYS of the three samples is remarkably increased. However, the increase in PS of the three samples is not obvious compared to that of the as-extruded sample, which of the free-end torsion sample has a slight decrease.

From Figure 8, it can be seen that all samples exhibit a concave-up flow curve in the early stages of compression deformation, which is a representative feature of extension twinning-dominated deformation [26]. This is because both the pre-tension and free-end torsion deformation cannot achieve full extension twinning deformation, i.e., the volume of extension twins is insufficient. The c-axis of most grains for the three types of samples is still perpendicular to the ED, which is favorable for extension twinning during compression along the ED. However, pre-deformation can generate significant dislocation and extension twins and lead to work hardening, which can cause an increase in the CRSS required to activate dislocation slips and extension twins. As a result, the CYS of the pre-deformed specimen was significantly increased when compared to that of the as-extruded specimen. In addition, extension twins and dislocations were already present in the pre-deformation specimens, acting as a strong barrier to dislocation movement and extension twinning, which can increase CYS. Compared to other samples, the pre-tension and then torsion sample underwent two steps of pre-deformation, and has the highest dislocations density and volume percentage of extension twins, thus exhibiting the highest CYS.

### 3.4. Micro-Hardness Variation during Pre-Tension and Free-End Torsion

Micro-hardness measurements were taken at five locations evenly distributed along the diameter of the polished cross-section. The micro-hardness distribution on the cross-section of various samples is shown in Figure 9. It can be seen that the micro-hardness of the as-extruded sample shows little variation from the center to edge position on the cross-section due to its homogeneous microstructure, while the micro-hardness values varied within the range of 52 HV to 53.5 HV. In contrast, the micro-hardness values of the pre-tension specimen are clearly increased, varying within the range of 63.5 HV to 65 HV. The micro-hardness values are distinctly increased because pre-tension deformation leads to work hardening. It is well known that the strengthening mechanism during plastic deformation in metallic materials is commonly attributed to dislocation strengthening and grain boundary strengthening [12]. Based on dislocation theory, dislocation density increases with the increase in plastic strain. Dislocation slip will become more and more difficult as the strain continues to increase [13]. As a result, pre-tension deformation becomes more and more difficult, leading to work hardening of the as-extruded AZ31 Mg alloys. This means that the micro-hardness increases with the increase in the pre-tension strain. However, the pre-tension deformation cannot refine grains, as shown in Figure 5. It can be concluded that the increase in the micro-hardness of the pre-tension specimen is mainly due to dislocation strengthening. Free-end torsion deformation as a plastic deformation mode that can also improve micro-hardness [21]. From the center to the edge of the cross-section, the micro-hardness values of the free-end torsional as-extruded specimen increases almost linearly, ranging from 53.5 HV to 67.5 HV. As the shear strain increases linearly during free-end torsion deformation, the micro-hardness values increase linearly from the center to the edge. This means that the center region suffers the lowest shear stress, resulting in the lowest dislocation storage and can achieve the lowest micro-hardness, while the edge region suffers the highest shear stress, resulting in the highest dislocation storage and can achieve the highest micro-hardness. Although the center position of the free-end torsional as-extruded samples contains a significant amount of extension twin lamellae, the micro-hardness in the center position is slightly increased when compared to the as-extruded sample, as shown in Figure 9. It can be concluded that grain refinement strengthening via extensional twin boundary subdivision is very limited on micro-hardness. In contrast, dislocation strengthening plays an important role in micro-hardness [21]. In our previous research, an annealing treatment was used to deal with the free-end-torsion-deformed AZ31 Mg alloy samples. The results showed that the annealing treatment can eliminate most of the dislocations, but retain the extension twins at 200 °C; furthermore, the micro-hardness of the annealed sample was decreased, proving that the extension twins have a smaller influence on the micro-hardness than that of dislocations [21]. This is because dislocations in metallic materials can be divided into statistically stored dislocations (SSDs) and geometrically necessary dislocations (GNDs) [34,35]. It is well known that SSDs grow in random trapping processes, while their density increases with an increase in plastic strain. Nevertheless, inhomogeneous plastic strain will lead to GNDs. Free-end torsion can give rise to non-uniformity shear strain on the cross-section, resulting in GNDs to accommodate the plastic deformation. As the strain gradient increases, the density of the GNDs also increases [13]. When shear strain increases, the strain gradient also increases, while the density of SSDs and GNDs further increases, which can lead to strengthening of the as-extruded AZ31Mg alloys. As a result, free-end torsion deformation can generate a gradient hardening effect on the cross-section, resulting in a gradient micro-hardness distribution on the cross-section of the free-end torsion sample.

From the center to the edge on the cross-section, the micro-hardness values of the pre-tension sample varied from 66.2 HV to 71 HV by free-end torsion. In comparison, the micro-hardness values of the as-extruded sample ranged from 53.5 HV to 67.5 HV by free-end torsion. The increase in micro-hardness of the as-extruded sample was approximately 14 HV, while that of the pre-tension sample was only about 4.8 HV. The reason for this is that the density of dislocations is affected by the formation and annihilation of dislocations during plastic deformation processes [13]. This means that, due to a balance between dislocation accumulation and dislocation annihilation, dislocation hardening will stabilize when the pre-tension specimen is subjected to free-end torsion. Consequently, the micro-hardness does not increase significantly and can even reach saturation.

In this study, the ATEX software was also used to calculate the density of GNDs from the EBSD data. The ATEX software (Version 2.08) can calculate the misorientation between each point within a selected region and a reference pattern, while the EBSD-GNDs maps are closely related to the local dislocation content [36]. Figure 10 displays the geometrically necessary dislocations and the corresponding distribution maps. It can be seen that the local GNDs in the edge area of the pre-tension and then torsion sample are relatively high in some grains, whereas they are relatively low in the initial specimen, as shown in Figure 10a,f. It can be concluded that the micro-hardness is closely correlated with the density of GNDs.

In previous work, it has been found that pre-compression has little effect on the torsion yield strength and reduces the strain hardening rate [37]. However, in this study, it was found that pre-tension can remarkably enhance the torsion yield strength and maintain a high strain hardening rate. This can be attributed to the different microstructure evolution between tension and compression. For compression, the generation of {10–12} twins will largely change the initial texture and generate a texture-softening effect [37]. However, pre-tension does not change the initial texture (see Figure 5) and introduces a high density of dislocations (see Figure 7 and Figure 10). A strong dislocation-hardening effect largely improves the torsion yield strength. Initial dislocations via pre-tension exhibit little influence on texture evolution during free-end torsion (see Figure 7); however, they might exacerbate the accumulation of dislocations (see Figure 10). Thus, combined use of pre-tension and free-end torsion can further enhance the compressive yield strength. Moreover, the combined use of pre-tension and free-end torsion can also further adjust the distribution of gradient microstructures (see Figure 7 and Figure 9), which provides a new approach for the preparation of high-performance Mg alloy bars.

## 4. Conclusions

In this work, the influence of pre-tension on free-end torsion deformation of an extruded AZ31 Mg alloy rod was studied. The microstructure evolution, mechanical properties and micro-hardness were investigated and discussed in detail. The main conclusions are as follows:

(1) Non-basal slip is the dominant deformation mechanism during pre-tension along the ED for the as-extruded AZ31 Mg alloy. Pre-tension can distinctly generate dislocation strengthening, which can increase the torsion yield strength and make the shear stress–shear strain curves of the pre-tension sample almost parallel to that of the as-extruded sample during the plastic deformation stage.

(2) Texture in the edge position on the cross-section for the two types of samples was rotated by about 30° along the ED. A great deal of extension twins occurred in the central area on the cross-section for the free-end torsion samples, and these were mainly due to the Swift effect. Meanwhile, extension twins that occurred in the edge area were mainly due to the normal stress during free-end torsion.

(3) For the free-end torsion specimens, the micro-hardness values increased almost linearly as the shear strain increased linearly from the center to the edge on the cross-section. However, the micro-hardness of the pre-tension and then torsion sample did not distinctly increase because the dislocation density became saturated during pre-tension deformation. Dislocation density was closely correlated with the micro-hardness. However, extension twins have a small effect on the micro-hardness.

## Figures and Tables

**Figure 1 materials-16-05343-f001:**
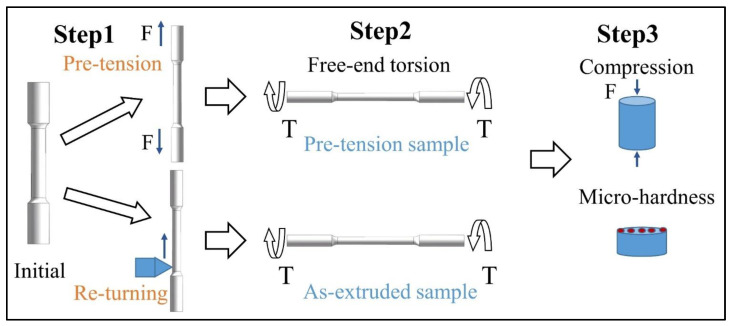
A schematic diagram of the present experiment.

**Figure 2 materials-16-05343-f002:**
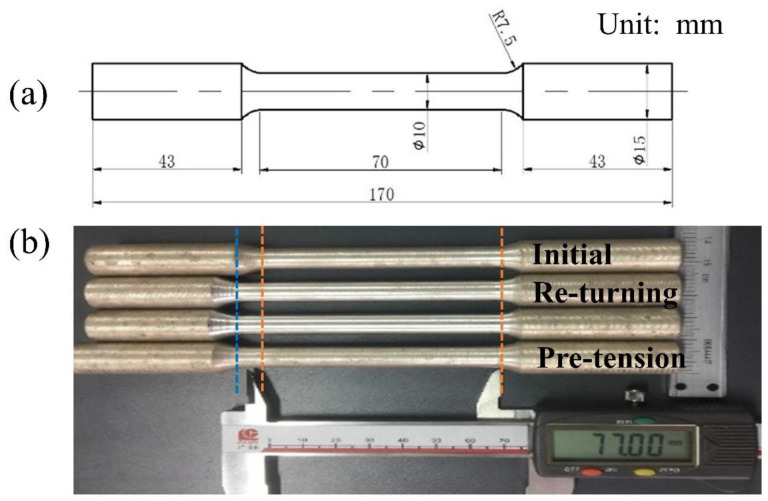
(**a**) The geometry and dimensions of the dog-bone-shaped sample, all dimensions are in mm; (**b**) photograph of the initial sample, the re-turning sample and pre-tension sample, respectively. The unit is in mm.

**Figure 3 materials-16-05343-f003:**
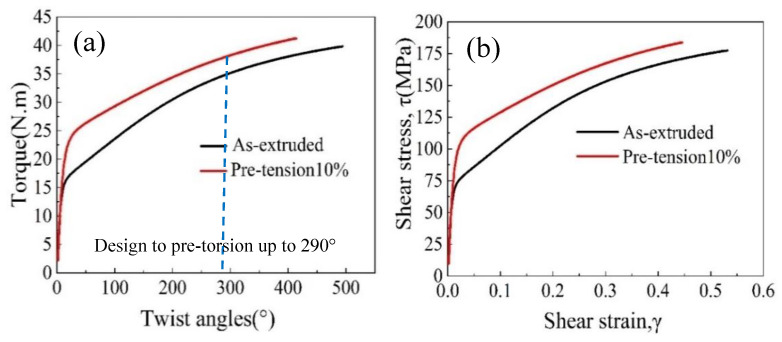
(**a**) Torque–twist angle curve, and (**b**) shear stress–shear strain curve obtained from the measured torque–twist angle curve.

**Figure 4 materials-16-05343-f004:**
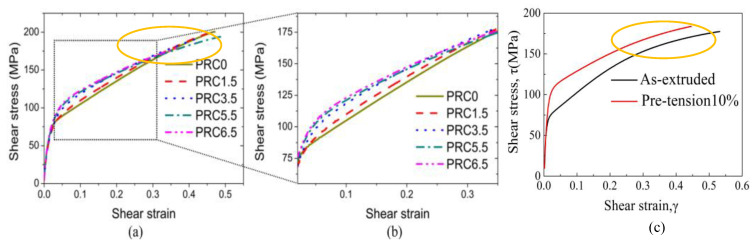
Shear stress–shear strain curve of the free-end torsion samples: (**a**,**b**) a commercially extruded AZ31 rod underwent pre-compression followed by free-end torsion in the literature [11]. A commercially extruded AZ31 (Mg-3 wt.%Al-1 wt.%Zn) Mg alloy with a diameter of 25 mm was used. The specimens were subjected to pre-compression up to the plastic strains of 0, 1.5%, 3.5%, 5.5% and 6.5% at a constant strain rate of 3 × 10^−4^/s, denoted as PRC0, PRC1.5, PRC3.5, PRC5.5, PRC6.5, respectively. After pre-compression, the specimens were subjected to free-end torsion at a constant rate of 5°/min. (**c**) As-extruded AZ31 rods underwent pre-tension followed by free-end torsion in the current experiment. The as-extruded alloy AZ31 (Mg-3 wt.%Al-1 wt.%Zn) with a diameter of 16 mm was used. The specimens were subjected to pre-tension up to a strain level of 10% at a constant strain rate of 1 × 10^−3^/s. After pre-tension, the specimens were subjected to free-end torsion at a constant rate of 180°/min.

**Figure 5 materials-16-05343-f005:**
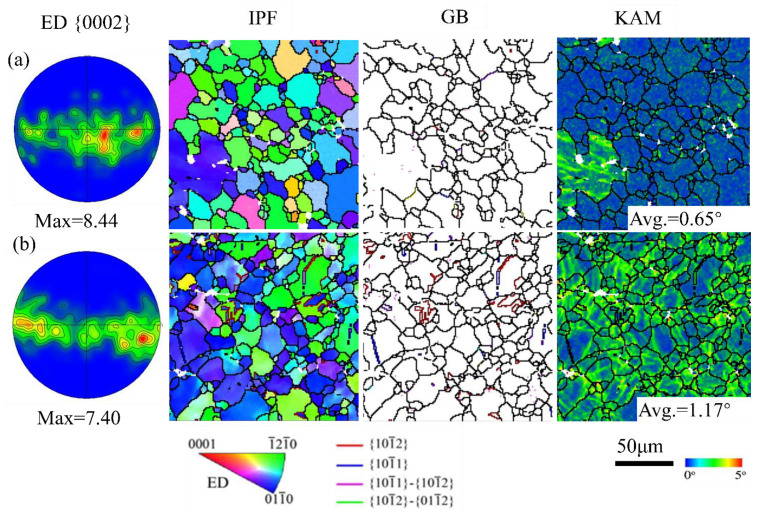
(0002) pole figure, EBSD inverse pole figure maps, grain and twin boundary maps and kernel average misorientation maps of (**a**) the as-extruded sample and (**b**) the pre-tension 10% sample.

**Figure 6 materials-16-05343-f006:**
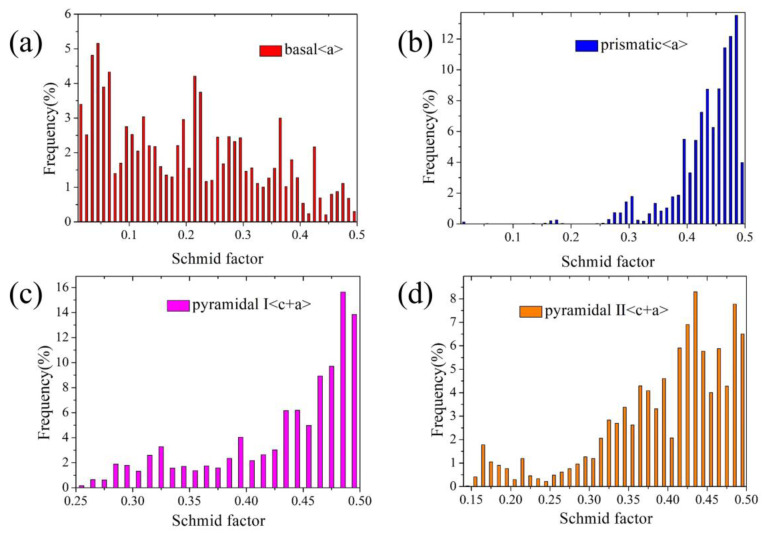
Schmid factor frequency distributions of (**a**) basal <a>, (**b**) prismatic <a>, (**c**) pyramidal I <c + a> and (**d**) pyramidal II <c + a> slips in the as-extruded sample, calculated from the tension direction and the IPF maps in Figure 5.

**Figure 7 materials-16-05343-f007:**
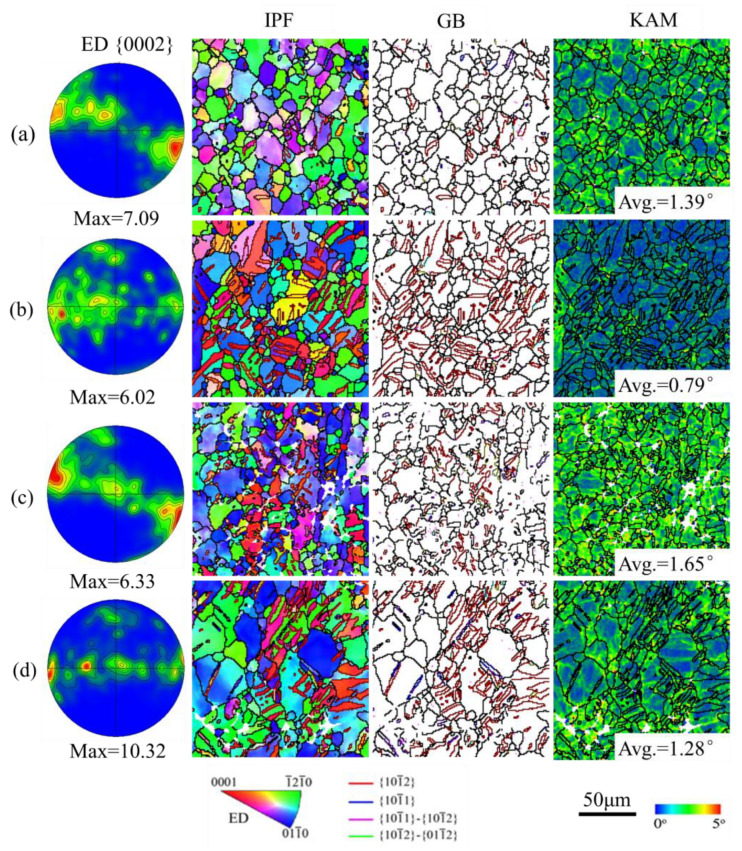
(0002) pole figure, EBSD inverse pole figure maps, grain and twin boundary maps and kernel average misorientation maps: (**a**,**b**) the edge and center area on the cross-section of the as-extruded sample was subjected to free-end torsion; (**c**,**d**) the edge and center area on the cross-section of the pre-tension sample was subjected to free-end torsion.

**Figure 8 materials-16-05343-f008:**
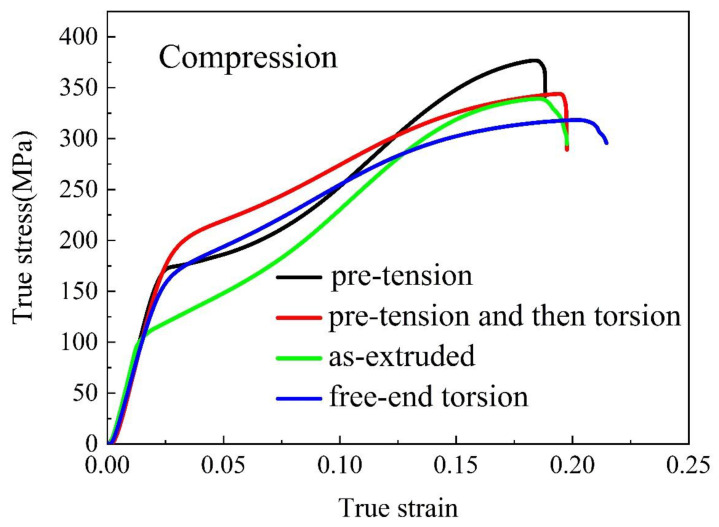
Compressive stress–strain curves of the various samples.

**Figure 9 materials-16-05343-f009:**
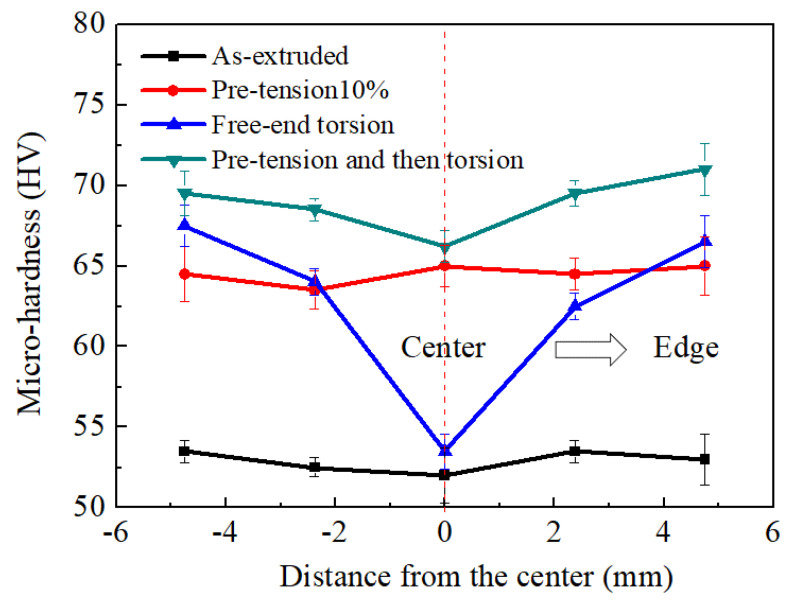
Micro-hardness at various radial positions of the different samples.

**Figure 10 materials-16-05343-f010:**
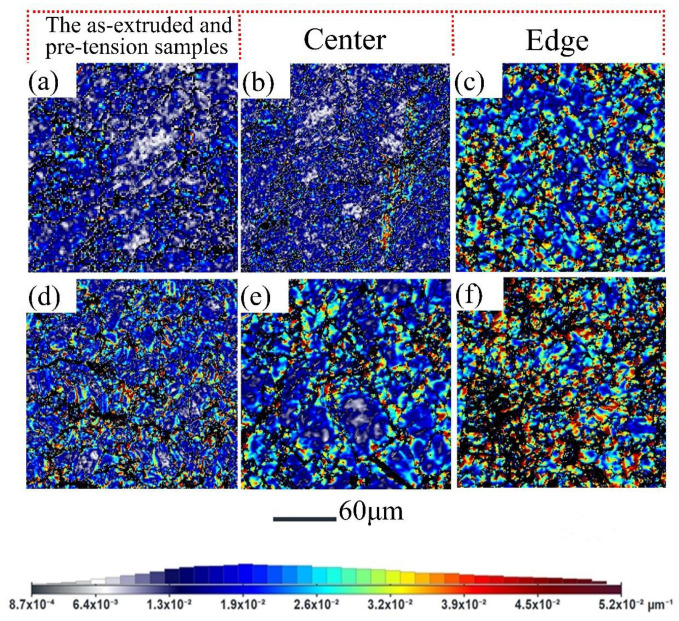
EBSD-GNDs density maps: (**a**) as-extruded sample, (**b**,**c**) the center and edge area on the cross-section of the as-extruded sample was subjected to free-end torsion, (**d**) pre-tension 10% sample, (**e**,**f**) the center and edge area on the cross-section of the pre-tension sample was subjected to free-end torsion.

**Table 1 materials-16-05343-t001:** Compressive mechanical properties of the different samples.

Mechanical Properties	As-Extruded	Pre-Tension	Free-End Torsion	Pre-Tension and then Torsion
CYS (MPa)	110.8 ± 3.5	174.9 ± 4.2	173.9 ± 2.7	206.8 ± 2.8
PS (MPa)	338.7 ± 1.3	377.8 ± 0.5	319.1 ± 2.1	342.3 ± 1.5
Percentage increase in the CYS (%)	_	57.85	56.95	86.64
Percentage increase in the PS (%)	_	11.54	−5.79	1.06

## Data Availability

All data generated or analyzed during this study are included in this published article.

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
