# Peer review of "Influence of Pre-Tension on Free-End Torsion Behavior and Mechanical Properties of an Extruded Magnesium Alloy"

_materials, 2023, doi:10.3390/ma16155343_

Round 1

Reviewer 1 Report

firstly, the title is not explicite enough to be undersood in considering the presented content. According to the present title, the torsion behavior would be studied, but in really, only the compression behaviour has been checked.

Secondly, in abstract, in introduction and in experiment part, it is necessary to justify the technique choice about the pre-tension, torsion and compression flowchart;

Thirdly, a separated discussion before conclusion should be added to leave a global synthesis about the different results and to explain the results evolution.

Furthermore, some experiment details should be added:

- the alloy composition and the corresponding heat-treatment conditions should be preicsed;

- in figure 2(a), the used unity should be indicated,

- the torsion test conditions should indicated explictely;

- for compression test, in considering the sample geometry, it is absolutely necessary to precise the test protocole and to discuss the results validity;

The results presentation should also be improved:

- in caption of figure 4(a) and (b), it is necessary to precise the different sample origins and their preparation conditions;

- in fiugre 6, the different colors for the different orientations are  readable. 

- in table 1 and in figure 9, it is necessary to add the test precison/error bars;

Reviewer 2 Report

The paper presents the results of research on the effect of initial stresses in extruded magnesium alloys subjected to torsional loads. A popular magnesium alloy of the AZ31 grade, intended for metal forming, was used for the tests. The work has been written in an understandable way and concerns the current problem related to the possibility of designing the mechanical properties of magnesium alloys.

More important notes to work:

1.       The presentation of the scientific purpose of the research and potential areas of application of the results, which were initially formulated in the final part of the first chapter, requires extension.

2.       How large the deformation values were obtained in the extruded semi-finished products from which the test samples were made?

3.       The scale in Figure 10 is illegible and requires improvement.

4.       Can the observed changes in the material have practical applications?

Reviewer 3 Report

The presented article is neatly written and well organized, and the results are discussed in detail. There are only a couple of minor comments.

1) The chemical composition of the studied alloy should be presented.

2) It is desirable to indicate the dimensions of the samples for compression tests.

Reviewer 4 Report

The article presents the results of research determining the influence of residual stresses on the free torsion of the extruded magnesium alloy AZ31. Magnesium alloys are currently widely used in the automotive and aviation industries mainly due to their low specific gravity, good properties and the prevalence of magnesium in the earth's crust. Testing the properties of magnesium alloys is therefore justified and of a utilitarian nature.

Overall, the article is understandable and well written.

The introduction sufficiently describes the current state of knowledge with references to references and justifies undertaking the research topic.

Materials and research methodology have been sufficiently described.

The test results were clearly presented. Their interpretation is not objectionable. Discussion of the obtained research results was carried out in relation to the existing state of knowledge with reference to References. Small note: The hardness measurement result should be the average value of several measurements and the measurement error should be stated.

Reviewer 5 Report

Please clarify what is ED in the abstract; better to avoid abbreviation.

Please introduce a schematic diagram so that the variables used in equation (1) can be understood clearly.

Round 2

Reviewer 1 Report

in the revised version of the manucript, author has made necessary corrections including all remarks from reviewers.